

# 'Unwilling' *versus* 'unable': Tonkean macaques' understanding of human goal-directed actions

Charlotte Canteloup[1,2,3,4,5] and  Hélène Meunier[1,2,3,4]

[1] Centre de Primatologie de l'Université de Strasbourg, Niederhausbergen, France
[2] Université de Strasbourg—Laboratoire de Neurosciences Cognitives et Adaptatives, Strasbourg, France
[3] CNRS—Laboratoire de Neurosciences Cognitives et Adaptatives, UMR 7364, Strasbourg, France
[4] University of Strasbourg Institute for Advanced Study, Strasbourg, France
[5] Department of Ecology and Evolution, University of Lausanne, Lausanne, Switzerland

## ABSTRACT

The present study investigated the understanding of goal-directed actions in Tonkean macaques (*Macaca tonkeana*) using the unwilling *versus* unable paradigm, previously used in several species. Subjects were tested in three experimental conditions that varied according to the goal-directed actions of a human actor. In the "unwilling" condition, the actor was capable of giving the subject food but unwilling to do it; in the "unable" condition, she was willing to give food but was unable to do it because of a physical barrier; and in the "distracted" condition, she was occupied by manipulating a pebble instead of food. We report for the first time that Tonkean macaques, like capuchins, chimpanzees and human infants, behaved differently across these experimental conditions. They attempted to grasp food in the actor's hand significantly more in the presence of an unwilling actor rather than an unable or a distracted one. Inversely, they begged significantly more facing a distracted and unable experimenter rather than an unwilling one. These results suggest that Tonkean macaques understand human goal-directed actions by predicting whether they were likely to obtain food merely based on movements, cue and motor intentions reading and understanding of physical constraints.

## INTRODUCTION

The perception of others as intentional agents is essential in human daily life and development. We attribute thoughts, beliefs and intentions to other people, and this helps us to better understand them, especially why they behave in certain ways, and to assess who are the best social partners (*Woodward, 2009*). Being able to understand intentions thus has many advantages in social life. Gauging the goals of others allows individuals to extract information from the environment and to anticipate the future behavior of others; for example, when facing a competitor in a novel situation (*Call & Tomasello, 2008*; *Call, 2009*). Are humans alone in being able to read the content of others'

Corresponding author
Charlotte Canteloup,
charlotte.canteloup@gmail.com

mind? Or do our closest relatives, the non-human primates, share this ability to represent other agents' mental states?

First, we need to define several terms such as 'goal-directed action', 'intentionality' and 'intention' that are commonly used in Theory of Mind research but rarely defined in the literature. On the one hand, a goal-directed behavior has been defined as "predominantly determined by endogenously generated (and internally represented) goals, rather than by external stimuli" (*Pezzulo & Castelfranchi, 2009*; p. 559). In this view, such action is expected to produce desired results—*goals*—and is guided toward these goals by the inter-play of prediction, control and monitoring. A goal-directed action would thus imply knowledge of the causal relationships between actions and their consequences, and a desire for the expected consequences or goal (*De Wit & Dickinson, 2009*). On the other hand, some authors consider goal-directed action as a particular relationship that animate agents have with objects and environmental states without postulating the existence of internal goals (*Penn & Povinelli, 2009*). In this view, nonhuman animals reason on the basis of perceptual similarity between a given situation and a past one by simply matching them, without reasoning in terms of causal mechanisms involving unobservable mental states. Philosophers of mind have defined intentionality as the property that makes all mental states and events directed toward, or relative to, objects or situations in the world (*Dennett, 1971*; *Searle, 1983*; *Brentano, 1995*). Intention has been defined as the "mental process of steering and controlling actions until the intended goal is achieved" (*Pezzulo & Castelfranchi, 2009*; p. 562) and as "a plan of action the organism chooses and commits itself to in pursuit of a goal" (*Tomasello & Carpenter, 2005*; p. 676). According to Buttelmann and collaborators (*2008a*), intentions comprised both a goal - *what* a person is doing—and a means chosen to achieve that goal - *how* she is doing it –and the rational choices of action plans—*why* she is doing it in that particular way. This is in accordance with the two levels of intentions proposed by philosophers: a first, behavioral level named 'intention in action' (*Searle, 1983*) or 'informative intention' (*Sperber & Wilson, 1995*), corresponding to the expression "I am doing X" (goal), and a second, psychological level labeled 'prior intention' (*Searle, 1983*) or 'communicative intention' (*Sperber & Wilson, 1995*), corresponding to the expression "I will do X" (rational choice). In this view, the first level of intention that can be directly perceived through bodily movements causes a second level of intention that can only be inferred. The intention is thus embedded in action.

The idea that intention is embedded in action is emphasized by the discover of mirror neurons in macaques' pre-motor cortex that discharge both when the monkey acts and observes a similar act done by another individual (e.g., *Gallese et al., 1996*). This discovery leads researchers to conclude that these parietal-frontal mirror neurons allow an observing individual to basically understand the goal of an observed action through behavior (e.g., *Rizzolatti, Fogassi & Gallese, 2001*; *Gallese, 2007*). According to these researchers, the monkey recognizes the goal of the motor act done by an observed individual because it knows the outcome of the act it executes. Moreover, *Fogassi et al. (2005)* showed that many motor inferior parietal lobule neurons fired during the observation of an act but also just before the beginning of the subsequent acts specifying the action, that shows that these

neurons code the observed motor act but also allow the observer to anticipate future acts and to understand then the agent's intentions.

Different methodologies have been used in experimental psychology and comparative ethology to study intention reading abilities. One method frequently used with human infants concerns imitation. In the Gergely et al. study (*Gergely, Bekkering & Király, 2002*), 14-month-old children watched an adult turn on a light with her forehead. For half of the infants, the adult was forced to use this unusual action because her hands were occupied; the other half of the infants saw the adult displaying the same action despite her hands being free. When given the unconstrained possibility to act on the light themselves, 69% of infants re-enacted the head action after watching the hands-free condition whereas only 21% of infants reproduced the action after watching the hands-occupied condition. The authors proposed that infants inferred that the head action offered some advantage if it was used even if the adult's hands were free. *Buttelmann et al. (2007)* found a similar effect in chimpanzees (*Pan troglodytes*), suggesting that infants and great apes understand the rationality of actions (but see *Buttelmann et al., 2013* for negative results concerning imitation in chimpanzees).

A second method often used to test infants and nonhuman primates' understanding of intentions is the accidental *versus* intentional protocol. *Carpenter, Akhtar & Tomasello (1998)* showed 14 to 18-month-old infants an adult demonstrating either an intentional action in which the adult exclaimed: ''There!'' or an accidental action in which the adult said: ''Whoops!''. Following the demonstrations, infants were given the opportunity to make the action themselves. The authors reported that infants imitated significantly more intentional than accidental actions, and concluded that they understood something about people's intentions. *Call & Tomasello (1998)* compared discrimination between a human's intentional and accidental actions in 2- and 3-year-old children, and older chimpanzees and orangutans (*Pongo pygmaeus*). Subjects learned to use the marker shown by the experimenter as a reliable cue to the location of a food reward in one of three boxes. In the experiment, the experimenter marked one box intentionally (by deliberating placing the marker) and one box accidentally (by accidentally dropping the marker). The subjects were then allowed to select one box. The results showed that the three species significantly selected the intentionally marked box more often than the accidentally marked one, suggesting shared sensitivity to the intentional nature of the experimenter's actions (*Call & Tomasello, 1998*). By contrast, *Povinelli et al. (1998)* found negative results using a similar paradigm in chimpanzees. Using a slightly different protocol, *Wood et al. (2007)* reported that chimpanzees, cotton-top tamarins (*Saguinus oedipus*) and rhesus macaques choose an intentionally targeted container more frequently than an accidentally marked one, and concluded that these species were able to infer rational and goal-directed actions of a human. Recently, the same paradigm has been applied to Tonkean macaques (*Macaca tonkeana*) and tufted capuchin monkeys (*Sapajus apella*) but with no evidence that these monkeys recognized others' goals (*Costes-Thiré et al., 2015*).

A third method used to test the attribution of intentions is the unwilling *versus* unable paradigm. In the original study using this paradigm (*Call et al., 2004*), after habitually feeding chimpanzees through a hole in a Plexiglass wall, the experimenter suddenly stopped

feeding them because either (i) he did not want to though he still could (unwilling), or (ii) he wanted to but could not (unable). The authors reported more spontaneous begging and auditory behaviors, and shorter latencies to leave by the chimpanzees when confronted with an unwilling compared with an unable experimenter, leading the authors to conclude that chimpanzees interpreted human actions as goal-directed. Similar results have been found in human infants from nine months of age (*Behne et al., 2005*), and in capuchin monkeys for actions displayed by a human but not those performed by inanimate rods (*Phillips et al., 2009*).

Despite differing views (*Lurz & Krachun, 2011*; *Povinelli & Vonk, 2003*), numerous researchers concluded that great apes can read below surface behavior to understand something about the goals, perceptions and intentions of others (*Tomasello & Carpenter, 2005*; *Tomasello et al., 2005*; *Call & Tomasello, 2008*; *Buttelmann, Call & Tomasello, 2008b*; *Buttelmann et al., 2012*). Studies on monkeys are fewer and evidence of Theory of Mind abilities as intention-reading abilities in these species remains scarce (e.g., *Barnes et al., 2008*; *Phillips et al., 2009*; *Drayton & Santos, 2014*; see *Cheney & Seyfarth, 1990*; *Povinelli, Parks & Novak, 1991*; *Kummer, Anzenberger & Hemelrijk, 1996* for negative results in macaques and *Drayton, Varman & Santos, 2016* for negative results in capuchins). From this perspective, we investigated understanding of goal-directed actions by adapting a protocol previously used with human infants (*Behne et al., 2005*), chimpanzees (*Call et al., 2004*), capuchins (*Phillips et al., 2009*) and African grey parrots (*Psittacus erithacus*: *Péron et al., 2010*) in a little known old world monkey species, the Tonkean macaque. The literature on this species, and notably on its social cognition, is indeed still scarce, in spite of its known very tolerant sociality. On the one hand, by testing a non-ape species, we aim at tracing back the evolutionary history of such characteristics in the primate lineage by bringing new data on a monkey species. On the other hand, and in a more general way, studying a tolerant species could bring new light on the effects of sociocultural environment on cognitive abilities. We tested macaques in three experimental conditions: (i) Unwilling: the experimenter did not want to give food to the subject; (ii) Unable: the experimenter could not give food to the subject because of a physical barrier and (iii) Distracted: the experimenter was manipulating a pebble instead of food. In case that Tonkean macaques lack the ability to understand human goal-directed actions, they would not discriminate between the three experimental conditions. Such results would question the idea that goal-directed actions understanding is a formerly shared trait with at least the common ancestor we have with macaques. However, if they are able to track the human goal-directed actions, as human infants and chimpanzees do, this ability could be shared by our common ancestor with macaques. In this way, we expected them to behave differently toward a human who was distracted, unwilling or unable to give them food. More precisely, we predicted that they would display more gaze alternation between the human and the food and more agonistic behaviors especially in the 'Unwilling' condition compared to the 'Unable' and 'Distracted' ones. Conversely, we hypothesized that macaques would be less attentive by looking more elsewhere facing a distracted or unable experimenter than an unwilling one.

## MATERIALS AND METHODS

### Ethical note

The procedures used here adhered to the EU Directive 2010/63/EU for animal experiments. This experiment was approved by the Animal Experiment Committee of the Centre de Primatologie de l'Université de Strasbourg and by the CREMEAS Ethics Committee (Approval for conducting experiments on primates no AL/46/53/02/13).

### Subjects

The subjects were fifteen Tonkean macaques (thirteen males aged 3–12 years and two females aged 6 and 16 years), all born and raised at the Centre de Primatologie de l'Université de Strasbourg, France. They lived in one of two groups: group A was composed of five adult males living in multi cage complex of two outdoor areas ($14.40 + 16.00$ m$^2$) connected to two indoor areas ($23.78 + 8.73$ m$^2$), and group B contained 26 individuals living in a 2,694 m$^2$ wooded park with access to a 20 m$^2$ indoor housing area. Subjects were tested between July and August 2015. Their daily diet consisted of commercial pellets and water *ad libitum,* and fruits and vegetables twice a week, out of experimental sessions.

### Apparatus

All the fifteen subjects were tested within their social group in an outdoor area situated alongside their indoor area for group A and alongside their park for group B. A concrete block ($58 \times 19$ cm) was placed inside the test area perpendicularly to the mesh, about 1 m from the ground, to be used as a seat by the subject (Fig. 1). In the experimenter area, a table ($85 \times 50$ cm) was placed in front of the subject. A horizontal opening ($64 \times 5$ cm) in the mesh allowed subjects to beg for food by extending their hand through the opening. Above the table, a Plexiglass panel ($100 \times 60$ cm) drilled with a feeding hole (3 cm in diameter, 22.5 cm above the horizontal opening) doubled the mesh on the experimenter's side. This small hole could be easily closed by a pivoting shutter ($10 \times 6$ cm).

### Experimental procedure

The experiment required one assistant and one experimenter (CC). Each subject participated in three different experimental conditions that varied according to the intentional action of the experimenter:

- Unwilling: the experimenter held a raisin and placed it near the edge of the table out of the subject's reach. She then grasped the raisin in one hand and made five explicit back and forth movements with this hand between her and the feeding hole. Each time her hand was close to the hole she made five small back and forth movements, preventing the subject from grabbing the raisin through the hole. The experimenter alternated her gaze between the raisin and the monkey's face.
- Unable: the experimenter held a raisin and placed it near the edge of the table out of the subject's reach. She then grasped the raisin in one hand and made five explicit back and forth movements with this hand between her and the feeding hole. However, prior to the trial, the assistant blocked the hole with the pivoting shutter. The experimenter's hand made the same back and forth movements as in Unwilling trials, but striking each

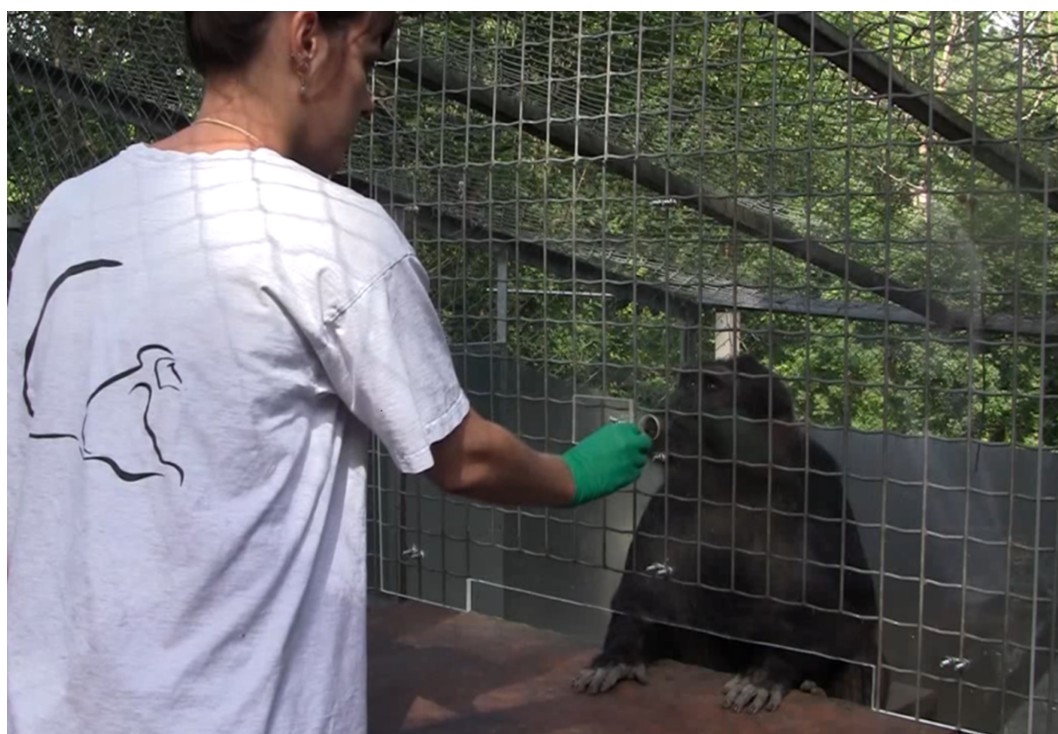

**Figure 1** Picture of the experimental apparatus during a trial of the 'Unwilling' condition.

time against the Plexiglass shutter. The experimenter alternated her gaze between the raisin and the monkey's face.

- Distracted: The experimenter held a raisin and a pebble and placed both on the table out of the subject's reach. She then grasped only the pebble in one hand, leaving the raisin on the table, and made five explicit back and forth movements with this between her and the feeding hole. Each time her hand was close to the hole she made five small back and forth movements. The experimenter alternated her gaze between the pebble and the monkey's face.

Six sessions per subject were conducted on a minimum of six different days. One session was composed of 12 trials including three experimental trials—one for each experimental condition—and nine motivational trials in which the food was placed on the table and then given to the subject through the feeding hole when the subject begged. Each experimental trial lasted about 30 sec and no food was given to the subject at the end of the trial. All subjects were tested using the same series of motivational and test trials and the order of presentation of experimental trials was pseudo-randomized across sessions, involving a session to always begin with two motivational trials in order to motivate the subject to stay for participating to the experiment. Experimental trials were separated by two or three motivational trials. The inter-trial interval was about 10 s during which the experimenter briefly left the experimenter area. In total, each subject was tested six times in each of the three experimental conditions that consisted in a total of 18 experimental trials per

individual. Subject participation was voluntary; they were free to leave the apparatus and testing area at any time.

## Data and reliability analyses

Experiments were recorded by a high-definition video camera (Legria HF S20; Canon, Tokyo, Japan). Videos were analyzed frame by frame (1 frame = 0.04s) by CC using the software *The Observer XT 10.1.548*. For the analysis, we recorded the following behaviors:

- Duration of presence of the subject on the seat
- Duration of gaze elsewhere
- Frequency of gaze alternations between the experimenter and the raisin on the table and between the experimenter and the item (raisin or pebble) in the hand
- Frequency of begging gestures, i.e., the subject extends an arm through the horizontal opening of the wire mesh. This was recorded when the extension of the gesture was at its peak just before the subject began to retract or lower his arm
- Duration of attempting to grasp the item
- Duration of threat towards the experimenter
- Duration of yawn and self-scratch.

For reliability analysis, a random 20% of trials were analyzed by a naïve observer using *The Observer*, with a tolerance window of 120 ms corresponding to 3 frames. Inter-observer agreement was excellent for the all the behaviors recorded: presence (Cohen's $\kappa = 0.86$), grasping attempt (Cohen's $\kappa = 0.90$), gaze elsewhere (Cohen's $\kappa = 0.89$), begging (Cohen's $\kappa = 0.90$), gaze alternation (Cohen's $\kappa = 0.88$), threat (Cohen's $\kappa = 0.94$) and yawn and self-scratch (Cohen's $\kappa = 0.92$).

## Statistical analysis

Two types of mathematical models were used to determine whether experimental conditions influenced behavioral measures. First, Generalized Linear Mixed Models (GLMMs) for count data (i.e., with a Poisson law distribution) were fitted to test which experimental condition influenced variables including begging gestures and gaze alternations between the experimenter and the item in the hand. Second, because experimental trials did not last exactly the same, we fitted Linear mixed–effects models (LME) for logit transformed proportion data in order to test which experimental conditions influenced continuous variables as the proportion of time spent in the following behaviors: item grasp attempt, gaze elsewhere and yawn and self-scratch. Normality of the residuals was assessed graphically (QQ-plots).

In each model, to deal with repeated measures, experimental condition ('unwilling'; 'unable'; 'distracted') was considered a fixed effect and subject identity was assessed as a random effect. Tukey corrections were applied when performing multiple comparison tests between experimental conditions. All models were performed with R 3.1.2's package lme4 (*Bates et al., 2015*), with alpha set at 0.050.

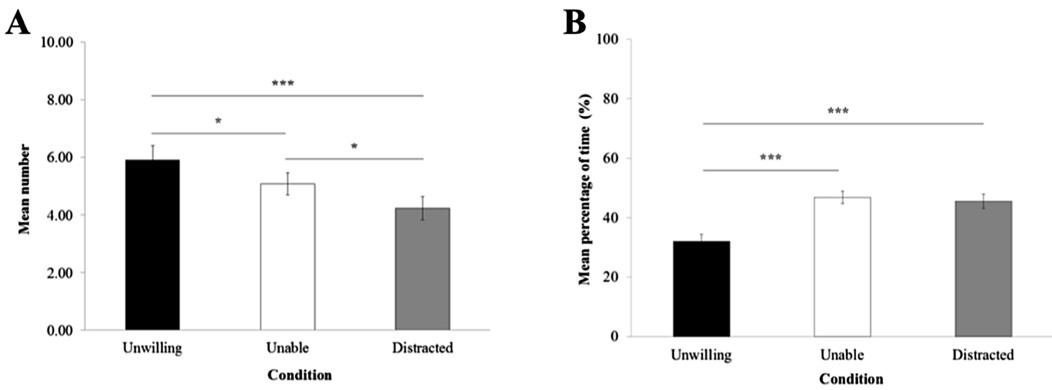

**Figure 2** **Gaze alternation and looking elsewhere.** (A) Mean number of gaze alternations between the experimenter and her hand holding the item per trial. (B) Mean percentage of time ($\pm$standard error of the mean) macaques looked elsewhere per trial.

## RESULTS

### Presence of the subject

Macaques spent more than 95 % of time on the seat in the three experimental conditions ('unwilling' condition: Mean percentage of presence time per trial $= 95.79\%$ $\pm$ Standard error of the mean $= 1.30$; 'distracted' condition: $95.36\%$ $\pm$ 1.65; 'unable' condition: $95.92\%$ $\pm$ 1.79).

### Gaze

The frequency of gaze alternations between the experimenter and the item in the experimenter's hand (Fig. 2A) was significantly influenced by the experimental condition ($LRT = 25.45$; $Df = 2$; $P < 0.0001$). GLMM revealed that macaques displayed significantly more gaze alternation in the 'unwilling' (Mean frequency per trial $\pm$ sem $= 5.91 \pm 0.49$) than the 'unable' condition ($5.08 \pm 0.39$; $P = 0.04$) and 'distracted' condition ($4.22 \pm 0.40$; $P < 0.001$). Also, more gaze alternations were detected in the 'unable' than 'distracted' condition ($P = 0.02$).

The proportion of looking time elsewhere (Fig. 2B) was significantly influenced by the experimental condition ($LRT = 35.352$; $Df = 2$; $P < 0.0001$). According to LME, macaques looked elsewhere for significantly longer in the 'distracted' condition compared with the 'unwilling' condition ($32.06\% \pm 2.37$; $P < 0.0001$), and in the 'unable' condition compared with the 'unwilling' condition ($P < 0.0001$). There were no significant differences between the 'unable' condition ($46.79\% \pm 2.04$) and the 'distracted' condition ($45.52\% \pm 2.41$; $P > 0.05$).

### Gestures

The frequency of begging (Fig. 3A) was significantly influenced by the experimental condition ($LRT = 129.15$; $Df = 2$; $P < 0.0001$). GLMM reported that macaques begged significantly more in the 'distracted' ($4.36 \pm 0.38$) than in the 'unable' ($2.43 \pm 0.26$)

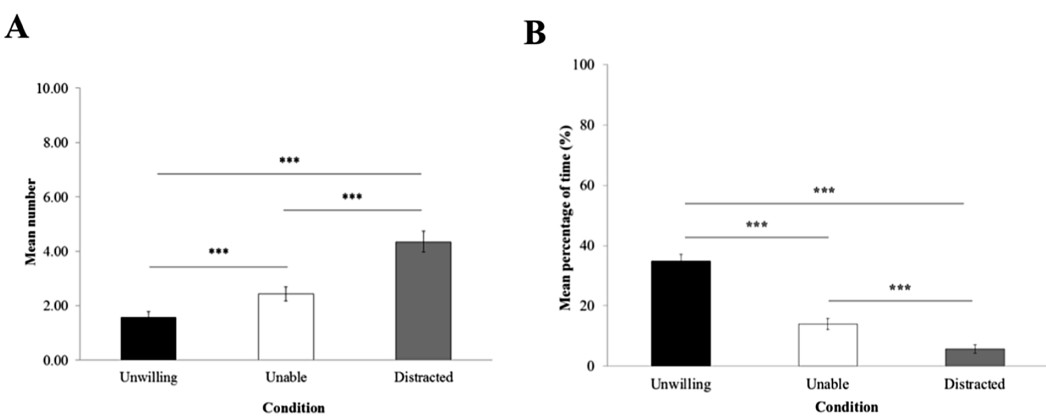

**Figure 3 Begging and grasping attempt.** (A) Mean number of begging gestures (±standard error of the mean) and (B) mean percentage of time of time (±standard error of the mean) macaques spent attempting to grasp the item in the experimenter's hand per trial.

and 'unwilling' conditions ($1.57 \pm 0.22$; $P < 0.0001$), and more in the 'unable' than the 'unwilling' condition ($P = 0.0002$).

The proportion of time trying to grasp the item through the hole (Fig. 3B) was significantly influenced by the experimental condition ($LRT = 129.93$; $Df = 2$; $P < 0.0001$). According to LME, macaques spent significantly more time attempting to grasp the item in the 'unwilling' condition ($34.78\% \pm 2.26$) than in the 'unable' ($13.94\% \pm 1.81$) and the 'distracted' conditions ($5.69\% \pm 1.41$; $P < 0.0001$). Also, macaques spent significantly more time trying to grab the item in the 'unable' than the 'distracted' condition ($P = 0.00066$).

### Threat, yawn and self-scratch

Only four individuals over fifteen displayed threat behaviors towards the experimenter at a very low level in the 'unwilling' ($0.48\% \pm 0.17$), 'unable' ($0.09\% \pm 0.09$) and the 'distracted' conditions ($0.02\% \pm 0.02$).

The proportion of time yawning and self-scratching (Fig. 4) was significantly influenced by the experimental condition ($LRT = 8.744$; $Df = 2$; $P = 0.012$). LME revealed significantly more time in these behaviors in the 'distracted' ($4.95\% \pm 1.01$) than the 'unwilling' condition ($2.33\% \pm 0.61$; $P = 0.015$). However, there were no significant differences neither between the 'unwilling' and the 'unable' conditions ($P > 0.05$) and between the 'unable' ($2.78\% \pm 0.72$; $P > 0.05$). Macaques tended to display more yawn and self-scratch behaviors in the 'distracted' than 'unable' condition ($P = 0.057$).

### DISCUSSION

We tested Tonkean macaques in the unwilling *versus* unable paradigm previously used in parrots (*Péron et al., 2010*), capuchins (*Phillips et al., 2009*), chimpanzees (*Call et al., 2004*) and human infants (*Behne et al., 2005*; *Marsh et al., 2010*). Like these species, Tonkean macaques behaved *as if* they understood the intentions of the experimenter, i.e., willing

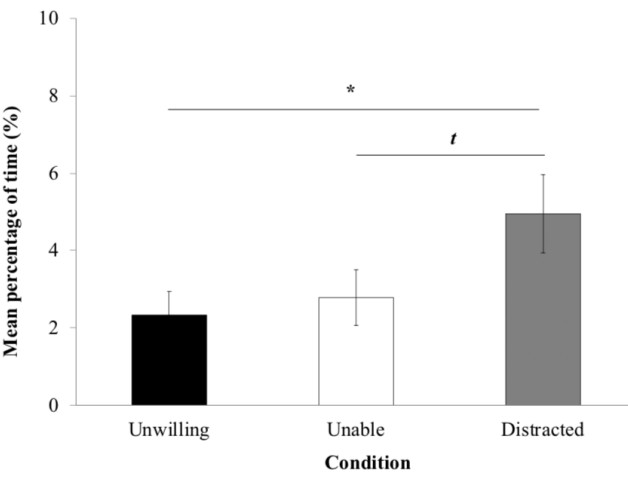

**Figure 4 Yawn and self-scratch.** Mean percentage of time macaques (±standard error of the mean) spent displaying yawn and self-scratch per trial.

to give them food or not, as they attempted to grasp the raisin in the experimenter's hand significantly more and were more attentive when she was unwilling rather than unable to give them food, or was distracted. We report for the first time that Tonkean macaques act differently according to the goal-directed actions of a human experimenter. Given that the experimenter displayed exactly the same gestural and visual behaviors in each experimental condition, our results cannot be explained by a low-level behavior reading. However, we cannot conclude from our experiment that Tonkean macaques truly understand all aspects of the underlying intentions of the human in a mentalistic way.

Tonkean macaques displayed significantly more gaze alternation between the experimenter's face and hand, and tried to grasp the item significantly more in the 'unwilling' than the 'unable' and 'distracted' conditions, and in the 'unable' than the 'distracted' condition. Moreover, Tonkean macaques spent more time looking elsewhere facing a distracted or unable experimenter than an unwilling one showing a disinterest for the experiment in these conditions. Together, these results indicate that, in accordance with results in human infants (*Behne et al., 2005*), chimpanzees (*Call et al., 2004*) and rhesus macaques (*Wood et al., 2007*), but in contradiction to a recent study in Tonkean macaques (*Costes-Thiré et al., 2015*), our subjects behaved differently according to experimental conditions corresponding to different goal-directed actions by a human experimenter. We can propose two explanations that could explain differences between *Costes-Thiré et al. (2015)* study and ours. On the one hand, the 'unwilling' *versus* 'unable' paradigm we used miss a communicative dimension: actions performed by the experimenter are not communicative and macaques do not need to understand the communicative intent of the action to understand the goal of the experimenter (e.g., I am going to obtain food). On the contrary, the 'accidental' *versus* 'intentional' paradigm used by *Costes-Thiré et al. (2015)* has a strong communicative component: subjects have to understand the experimenter's communicative intentions (e.g., She is trying to show me where the food is) to succeed the

experiment. That makes the task cognitively more demanding for macaques than simply inferring the experimenter's action goal. On the other hand, our Tonkean macaques had received no training prior to the experiment, as in other studies reporting positive results (*Behne et al., 2005*; *Call et al., 2004*; *Marsh et al., 2010*; *Phillips et al., 2009*; *Wood et al., 2007*), and unlike the study reporting negative results (*Costes-Thiré et al., 2015*). Moreover, it would be necessary to test our subjects in many more trials to observe a learning effect; chimpanzees needed hundreds of trials to discriminate between a human that could and could not see them (*Povinelli & Eddy, 1996*). Overall, future investigation of a larger sample of individuals would be desirable to strengthen the validity of our results and to examine more precisely potential learning effect. It is important to stress that, the experimenter acted in exactly the same way in the three experimental conditions in terms of gaze alternation and manual movements. From this perspective, our results cannot be explained by recourse to low-level behavior-reading based on the topography of the experimenter's motoric and visual behavior. By contrast, skeptics could propose that the macaques' grasping attempt behavior might simply reflect frustration at not receiving food. Indeed, we reported that Tonkean macaques attempted significantly more to grasp food when the experimenter was unwilling to give them food than when she was unable or distracted to do so. This result can be interpreted as a result of frustration of not obtaining the raisin that is close to reach but also as an understanding of experimenter's goal-directed actions. To rule out this explanation in their study of chimpanzees, *Call et al. (2004)* ran a non-social control condition in which the experimenter left the testing area after placing the food on the platform. In this condition chimpanzees produced fewer behaviors and left the testing area earlier compared to conditions in which he remained. On the one hand, we recognize that we did not run such a non-social control, but we previously reported in a comparable non-social condition that Tonkean macaques and rhesus macaques produced gestures intentionally towards a human experimenter and pointed significantly less towards food when the experimenter was absent (*Canteloup, Bovet & Meunier, 2015a*; *Canteloup, Bovet & Meunier, 2015b*) that makes then this explanation unlikely. On the other hand, another way to test for the frustration hypothesis would be to analyze results of agonistic behaviors and frustration behaviors displayed by macaques as yawning and self-scratching (*Maestripieri et al., 1992*). Subjects displayed significantly more frustration behaviors in the 'distracted' condition than the 'unwilling' one, and they tended to show more such behaviors facing a distracted experimenter rather than an unable one. This could be interpreted as a frustration of not obtaining the food visible on the table but out of reach in the 'distracted' condition compared to the other ones. Despite the fact that very few threats have been recorded and that no statistical analyses could then be run, it is nonetheless interesting to notice that, when threats occurred, it happened more facing an unwilling experimenter rather than an unable or distracted one. However, more data are necessary to strengthen the explanation that Tonkean macaques perceive the goals of the human actions.

The Tonkean macaques begged significantly more through the horizontal opening when the experimenter was distracted rather than when she was unwilling or unable to give them food, and more when she was unable than unwilling to give them food. The

greater incidence of begging in the 'distracted' condition compared with the others might be related to the raisin being out of reach on the table in this condition, eliciting attempts to grasp it or to attract the experimenter's attention towards the food. It appears thus clear that the macaques understood that the Plexiglass panel was a physical barrier in the 'unable' condition, making the transfer of food impossible. Begging would thus be an alternative way to attempt to obtain food from a well-intentioned experimenter. These results support the idea that Tonkean macaques understood that the physical barrier impeded the transfer of food in the 'unable' condition, and that they tried to solve the problem by raising their arm above the opening.

Contrary to capuchin monkeys (*Phillips et al., 2009*) and chimpanzees (*Call et al., 2004*), Tonkean macaques did not leave the testing area earlier when faced with an unwilling experimenter. According to those authors, capuchins and chimpanzees appear sensitive to the experimenter's intentions when determining how long to wait for food. However, Tonkean macaques remained present for more than 95 percent of time in the three experimental conditions. The fact that Tonkean macaques are a highly tolerant macaque species (*Thierry, 2000*) could explain why they were so patient, quiet and peaceful throughout the experiment, in comparison with species more despotic as chimpanzees. Simple "presence" thus does not appear to be a useful measure of discrimination of intentional actions in this species. Their social tolerance could also explain the low rates of threat in the 'unwilling' condition.

To sum up, our results add to previous ones reporting that, like great apes, some monkeys species seem also capable of estimating visual perception of others (*Flombaum & Santos, 2005*; *Overduin-de Vries, Spruijt & Sterck, 2014*; *Canteloup, Bovet & Meunier, 2015a*; *Canteloup, Bovet & Meunier, 2015b*; *Canteloup et al., 2016*) and the intentional nature of an action (*Call et al., 2004*; *Phillips et al., 2009*; *Wood et al., 2007*). *Schmitt, Pankau & Fischer (2012)* revealed using Primate Cognition Test Battery (see *Herrmann et al., 2007* for the initial test in human infants and chimpanzees) that monkeys were not outperformed by apes. Long-tailed macaques and baboons performed even better than chimpanzees and orangutans in some of the social cognitive tests. Our ability to assess mental states of others would not have appeared de novo but would rather be deeply tied to the evolutionary roots we share with our closest relatives the nonhuman primates. Rochat and collaborators (*2008*) reported that macaques monkeys, as human infants, looked longer at indirect events, indicating surprise for an unnecessary action and an understanding of the goal-directedness of actions. In this line, we suggest that Tonkean macaques understand goal-directed actions by perceiving a first level of intention labeled 'intention in action' (*Searle, 1983*) or 'informative intention' (*Sperber & Wilson, 1995*) in the literature, concepts that are directly perceivable through bodily movements. Indeed, intention is not a unitary concept but a multi-level one, and evaluation of an individual's action differs from the understanding of the individual's intentions (*Call & Tomasello, 2008*). Tonkean macaques seem thus able to understand intentional actions as pursuing goals persistently. According to *Povinelli & Vonk (2003)* and their 'behavioral abstraction hypothesis', macaques would form an association between the experimenter's behavior (food in hand close to me *versus* food far from me) and the outcome (obtaining food probable *versus* obtaining

food improbable). They may learn the rule: <*when there is a physical barrier between me and food, I cannot have access to food*>, and not have mentalized: <*the experimenter is well intentioned when trying to give me food but unable because of the physical barrier*>. *Phillips et al. (2009)* proposed another explanation of their results with capuchins: the monkeys might have a set of mechanical principles in mind construing that animate agents can move on their own, contrary to inanimate objects. This proposition is quite different from the 'teleological stance' adopted by *Gergely & Csibra (2003)*, in which interpreting goal-directed actions relies on the understanding of efficient action and physical efficiency of actions of both animate and inanimate agents. Our results fit with several theories as embodied social cognition proposing that cognitive processes operate on perceptual input and involve motor representations rather than representation of unobservable mental states (e.g., *Fenici, 2012*; *Gómez, 1999*).

To conclude, we reported that Tonkean macaques behaved *as if* they understood the actions and the underlying intentions of an experimenter. Despite the existence of high-level mindreading explanations (*Call & Tomasello, 2008*; *Dennett, 1971*; *Dennett, 1987*), all the existing findings and ours can also be explained by lower-level explanations whose behavior-reading hypotheses (*Butterfill & Apperly, 2013*; *Fenici, 2012*; *Gergely & Csibra, 2003*; *Heyes, 2014*; *Povinelli & Vonk, 2003*; see also *Meunier, in press* for a review). However, it may be worth re-evaluating the dichotomy between high *versus* low mental levels. For example, following *Grandin (1995)*; *Grandin (2002)*; *Grandin & Johnson (2004)*; *Grandin (2009)*, it might be proposed that nonhuman animals can develop internal representation of what others see, do etc. not in a language-based way, as humans do, but rather in a sensory-based way. From this perspective, animals develop a large visual or other sensory-based data bank in their brain that enables them to project their own experience to others, to take the visual perspective of others or to discriminate their intentions, not by thinking in words like normal humans do but rather by thinking in images, similar to what some autistic persons do (*Grandin, 1995*; *Grandin, 2009*). We should reconsider what ''mentalistic'' means: is it only linked to human language? Or does a different way of mentalizing exist, especially for creatures that have no spoken language? These questions may be valuable for framing future projects by researchers from the Humanities and Biological Sciences.

## ACKNOWLEDGEMENTS

The authors are grateful to Yves Larmet and the whole team of the Centre de Primatologie de l'Université de Strasbourg for allowing them to run this study. The authors are particularly thankful to Steve Lapp and Adrien Panter for their help in building the experimental apparatus. Lucie Hoornaert, Justine Guillaumont and Lena Buscara are greatly thanked for their assistance during the experiment. The authors would also like to especially thank Gaël Raimbault, who served as a second coder for the reliability analysis. Nicolas Poulin from Strasbourg University and Frédéric Schütz from Lausanne University are warmly thanked for their precious statistical assistance. James R. Anderson is warmly thanked for insightful discussions and for editing the English of the manuscript. The authors are also greatly

thankful to Luciano Fadiga, Academic Editor of PeerJ and two anonymous reviewers for their help in improving the paper.

### Funding
This study was funded by the University of Strasbourg Institute for Advanced Study (USIAS). The funders had no role in study design, data collection and analysis, decision to publish, or preparation of the manuscript.

### Grant Disclosures
The following grant information was disclosed by the authors:
University of Strasbourg Institute for Advanced Study (USIAS).

### Competing Interests
The authors declare there are no competing interests.

### Author Contributions
- Charlotte Canteloup conceived and designed the experiments, performed the experiments, analyzed the data, contributed reagents/materials/analysis tools, wrote the paper, prepared figures and/or tables, reviewed drafts of the paper.
- Hélène Meunier conceived and designed the experiments, contributed reagents/materials/analysis tools, reviewed drafts of the paper.

### Animal Ethics
The following information was supplied relating to ethical approvals (i.e., approving body and any reference numbers):

The procedures used here adhered to the EU Directive 2010/63/EU for animal experiments. This experiment was approved by the Animal Experiment Committee of the Centre de Primatologie de l'Université de Strasbourg and by the CREMEAS Ethics Committee (Approval for conducting experiments on primates no AL/46/53/02/13).

### Data Availability
The raw data has been supplied as a Data S1.

### Supplemental Information
Supplemental information for this article can be found online at http://dx.doi.org/10.7717/peerj.3227#supplemental-information.

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
