# Peer review of "‘Unwilling’ versus ‘unable’: Tonkean macaques’ understanding of human goal-directed actions"

_PeerJ, doi:10.7717/peerj.3227_

## Round 0.1 · original submission · Minor Revisions

Please, attentively address all the issues raised by the reviewers.

Reviewer 1 ·

Basic reporting

This research paper by Canteloup and Meunier describes the behavior of Tonkean macaques facing an unwilling, unable or distracted experimenter manipulating a food item. The results are in favor of a basic understanding of the experimenter’s intentions based on movements and physical constraints. This work presents a useful contribution to the field. It is overall well-written and displays clear and consistent results.

Below are my comments to improve the manuscript:

The introduction should be more straightforward. Some information presented is not crucial to the understanding of the study. Key definitions, important conclusions from previous studies and presentation of the original "unwilling vs unable" paradigm would be sufficient.

In introduction/discussion, a reference to previous neurophysiological studies addressing the question of goal/intention understanding, would be valuable (ex: Fogassi, Ferrari, Gesierich, Rozzi, Chersi and Rizzolatti. Parietal lobe: From action organization to intention understanding. Science 2005).
Previous work by Schmitt, Pankau and Fisher (Old World monkeys compare to apes in the primate cognition test battery. Plos One, 2012) would also be worth discussing with regard to social cognition in long-tailed macaques and olive baboons (communication, theory of mind).

Line 183-184: “...by filling the gap between chimpanzees and capuchins.”: I would omit this expression. Capuchins are highly specialized in terms of cognitive skills and manual dexterity and are not so representative of general skills of New World monkeys. As such, it is uncertain that a common trait between macaques and capuchins would trace back to the common ancestor. See also: Padberg et al. 2007. J. Neurosci. Parallel evolution of cortical areas involved in skilled hand use.

Line 193: “...displayed...” should be changed to “...display...”.

Line 241: “...close the hole...” should be changed to “...close to the hole...”.

Line 257: “...on a minimum six different days...” should be “on a minimum of six different days...”.

Experimental design

I have no comment on the experimental design. The research question is well defined and the methods are described with sufficient detail.

Validity of the findings

As I understood, 15 monkeys were tested in 3 different conditions, with 6 trials per condition. Although the small number of trials might have been necessary to avoid learning or habituation, it might be worth stating in discussion that future investigation of a larger sample would be profitable to better validate the present findings and to examine learning/habituation effects.

Overall, the results would be easier to understand and put in relation with the figures by starting from the greatest value. For example, lines 322-325 would be changed to: “...macaques displayed significantly more gaze alternation in the ‘unwilling’ (Mean ± SEM = 5.91 ± 0.49) than the ‘unable’ (5.08 ± 0.39; P = 0.04) and ‘distracted’ conditions (4.22 ± 0.40; P < 0.001). Also, more gaze alternations were detected in the ‘unable’ than ‘distracted’ condition (P = 0.02).”

Reviewer 2 ·

Basic reporting

Please see my comment below regarding the authors' directed hypothesis.

Experimental design

no comment

Validity of the findings

no comment

Additional comments

The authors presented Tonkean macaques with the unwilling-versus-unable paradigm and found differences between the two conditions (and a third, control condition) in a variety of subjects’ behaviors. They interpret these differences in performance as evidence for goal understanding in Tonkean macaques.
I need to say that I reviewed a previous version of this manuscript for another journal. This gives me the possibility to evaluate how the authors incorporated my suggestions and how they attempted to improve their manuscript in general. All in all, the authors did a very good job. They incorporated most of my previous comments and also changed other things (possibly due to feedback from others). These changes increased the readability of the paper and help to highlight the merit of this study.
The only issue that still feels surprising to me when reading the manuscript is the authors’ hypothesis that their subjects would succeed in the task (lines 189 – 196). Given Costes-Thiré, Levé, et al.’s (2015) results one might expect the opposite. What is the authors’ basis for their directed hypothesis? Do they believe the unwilling-versus-unable paradigm is more likely to worm the sophisticated social-cognitive abilities out of the monkey subjects than is the accidental-versus-intentional paradigm? If yes, why? They seem to suggest that a main difference between their study and that of Costes-Thiré is that in the current study subjects were not trained (line 394 – 397).
A possible reason for subjects’ success in the current task that came to my mind might be that in the unwilling-versus-unable paradigm the target actions performed by the experimenter are not communicative at all. In the Costes-Thiré et al. (2015) study subjects had to understand the experimenter’s communicative intentions (e.g., she is trying to show me where the food is). This seems significantly harder (given nonhuman animals’ competitive nature) and less straightforward than inferring an experimenter’s action goal (with no communicative intent being present).
Apart from this single issue, I think this manuscript is ready for publication. It will be of great interest to researchers in the fields of developmental and comparative psychology, biology and even philosophy.

Minor comment:
The reference “Drayton, Varman, & Santos, 2015” is cited in the text but cannot be found in the reference list. Please add it.

---

## Round 0.2 · Minor Revisions

Please, address reviewer's 2 concern.

Reviewer 1 ·

Basic reporting

No comment.

Experimental design

No comment.

Validity of the findings

No comment.

Additional comments

The authors answered all my comments from the first review. Results are well stated and the research question is clearly linked with the literature.

Reviewer 2 ·

Basic reporting

Meets all criteria.

Experimental design

Meets all criteria.

Validity of the findings

All criteria met.

Additional comments

I am pleased that the authors appreciated and included my idea about a possible reason for subjects’ success in the current task (in contrast to the Costes-Thiré et al., 2015, study). However, they did not address my main (and only) issue which was that there is no ground for a directed hypothesis here. In my opinion, one would have to to present both possible outcomes and say something like "In case Tonkean monkeys lack the abilitiy to ... . However, if they do track the goals of others in a way human infants and chimapnzees do, we expected them to ...". This would highlight the explorative character of the study... and it does not discredit the current study in any way.

---

## Round 0.3 · accepted · Accept

This is a very interesting contribution. I am happy to see that the review process has really improved the overall quality thanks to both, reviewers and authors.

Reviewer 2 ·

Basic reporting

no comment

Experimental design

no comment

Validity of the findings

no comment

Additional comments

This version looks perfect. The authors addressed all my comments accordingly, and I am happy to suggest publication of this interesting study.